# Interspecific Host Variation and Biotic Interactions Drive Pathogen Community Assembly in Chinese Bumblebees

**DOI:** 10.3390/insects14110887

**Published:** 2023-11-17

**Authors:** Huanhuan Chen, Guangshuo Zhang, Guiling Ding, Jiaxing Huang, Hong Zhang, Mayra C. Vidal, Richard T. Corlett, Cong Liu, Jiandong An

**Affiliations:** 1State Key Laboratory of Resource Insects, Key Laboratory of Insect-Pollinator Biology of Ministry of Agriculture and Rural Affairs, Institute of Apicultural Research, Chinese Academy of Agricultural Sciences, Beijing 100193, China; chenhuanhuan@caas.cn (H.C.); zhangguangshuo77@126.com (G.Z.); dingguiling@caas.cn (G.D.); huangjiaxing@caas.cn (J.H.); zhanghong@caas.cn (H.Z.); 2Centre for Yunnan Plateau Biological Resources Protection and Utilization, College of Biological Resource and Food Engineering, Qujing Normal University, Qujing 655011, China; 3Biology Department, University of Massachusetts, Boston, MA 02125, USA; mayra.cadorinvidal@umb.edu; 4Center for Integrative Conservation, Xishuangbanna Tropical Botanical Garden, Chinese Academy of Sciences, Mengla 666303, China; corlett@xtbg.org.cn; 5Department of Organismic and Evolutional Biology, Museum of Comparative Zoology, Harvard University, Cambridge, MA 02138, USA

**Keywords:** *Bombus*, China, coinfection, community assembly, host–pathogen interactions

## Abstract

**Simple Summary:**

Our study investigated the dynamics of pathogen communities within Chinese bumblebee populations, a key issue for understanding emerging infectious diseases and their impacts on insect biodiversity loss. By employing computational modeling on extensive pathogen data from bumblebees, we uncover that the host species variation significantly influences pathogen assembly and occurrences, compared to environmental factors like climate and location. Moreover, our results reveal significant pathogen–pathogen interactions, with similar pathogens exhibiting facilitatory relationships and distinct pathogen types showing strong negative associations, possibly due to immune response interactions and competitive dynamics within hosts. Our findings highlight the role of host–pathogen coevolution and other ecological interactions in shaping pathogen communities. The implications of this work are substantial for bumblebee conservation, improving our understanding of pathogen dynamics that could inform strategies to stop declines in bumblebee populations.

**Abstract:**

Bumblebees have been considered one of the most important pollinators on the planet. However, recent reports of bumblebee decline have raised concern about a significant threat to ecosystem stability. Infectious diseases caused by multiple pathogen infections have been increasingly recognized as an important mechanism behind this decline worldwide. Understanding the determining factors that influence the assembly and composition of pathogen communities among bumblebees can provide important implications for predicting infectious disease dynamics and making effective conservation policies. Here, we study the relative importance of biotic interactions versus interspecific host resistance in shaping the pathogen community composition of bumblebees in China. We first conducted a comprehensive survey of 13 pathogens from 22 bumblebee species across China. We then applied joint species distribution modeling to assess the determinants of pathogen community composition and examine the presence and strength of pathogen–pathogen associations. We found that host species explained most of the variations in pathogen occurrences and composition, suggesting that host specificity was the most important variable in predicting pathogen occurrences and community composition in bumblebees. Moreover, we detected both positive and negative associations among pathogens, indicating the role of competition and facilitation among pathogens in determining pathogen community assembly. Our research demonstrates the power of a pluralistic framework integrating field survey of bumblebee pathogens with community ecology frameworks to understand the underlying mechanisms of pathogen community assembly.

## 1. Introduction

Bumblebees (genus: *Bombus*) are among the most important pollinators on the planet, playing a vital role in maintaining biodiversity, the health of natural ecosystems, and agricultural food production [1]. However, recent reports of significant declines in insect abundance and biodiversity have raised global attention over the decline of bumblebees [2,3,4,5,6,7,8,9]. There are multiple reports of bumblebee declines in Asia [10,11], Europe [12,13,14], North America [15,16,17,18], and South America [19,20]. These declines have been attributed to several well-known causes, including natural habitat loss and fragmentation [21], global climate change [22], pesticide use [23], and invasion of non-native species [24]. However, a study on the rapid decline in the abundance of North American bumblebees has shown that invasive and emergent pathogens could significantly contribute to the global decline of wild bumblebee populations [17], leading to a growing concern about a more general phenomenon in conservation biology: the effect of emerging infectious diseases on wild populations in biodiversity and ecosystem services.

Although the exact number of pathogens affecting bumblebees is still being researched, several key classes of pathogens are known to afflict them. These include viruses, bacteria, fungi, and protozoans [1]. In recent years, the list of pathogens found in bumblebees has increased significantly [1], with several well-studied pathogens being strongly linked to the decline of bumblebee populations, including trypanosoma (e.g., *Crithidia* [25,26]), microsporidia (e.g., *Nosema* and *Apicystis* [27]), and viruses (e.g., acute bee paralysis virus and deformed wing virus (DWV) [28,29]). The impact of these pathogens on bumblebee health is significant. They can cause various effects, from sub-lethal symptoms that reduce foraging efficiency and colony productivity to lethal effects that can lead to the collapse of entire colonies [1]. Pathogens like DWV can cause developmental defects [28], while gut parasites like *Nosema* can affect their digestive system, leading to malnutrition and a weakened state [27]. Protozoan parasites such as *Crithidia* can impair the immune system and reduce lifespan [25]. Although most of these studies have either simply reported the list of pathogens found in bumblebees or evaluated the negative effect of a single pathogen, it is increasingly evident that bumblebees frequently harbor infections from multiple pathogens simultaneously [1]. Those multiple infections can exacerbate the overall health impact on bumblebees [1]. However, the processes and determinants that drive the species composition and community assembly of bumblebee pathogen communities are not yet well documented and understood. Understanding those processes has important implications for us to predict infectious disease dynamics and evaluate the cost of infections for bumblebee populations.

In the context of community ecology concepts and frameworks, host bumblebee species can be considered an empty patch with resources, and the composition of pathogen communities within a host is determined by stochastic factors, such as ecological drift [30], or more deterministic processes [31]. In the former view, the differences among pathogens are not important in determining which pathogen occupies which host individual, leading to random pathogen composition in bumblebee species. Under the deterministic scenario, environmental filtering (host resistance) could structure pathogen communities [31]. Bumblebee hosts have resistance against pathogen species, and the interspecific variation in their resistance may act as filters for selecting different subsets of pathogens, leading to non-random patterns of pathogen co-occurrence among bumblebee species. Another deterministic hypothesis suggests that the interaction dynamics between bumblebee pathogens play a significant role in shaping pathogen communities [32]. On the one hand, pathogen species coexisting within the same host may engage in competition for host resources, leading to negative species associations, with this competition expected to be stronger between more similar pathogens. On the other hand, pathogen species associations may be mediated by host immune responses. For example, early-arriving pathogens can suppress the establishment of late-arriving pathogens through cross-reactive immune responses, resulting in negative pathogen associations. Moreover, early-arriving pathogens may facilitate the transmission of other pathogens through immunosuppression.

Here, we study the relative importance of biotic interactions versus interspecific host resistance in shaping the pathogen community composition of bumblebees in China. China is a hotspot for bumblebee diversity, harboring over half of the world’s bumblebee species [33,34]. Yet, despite this rich diversity, studies on the pathogens of Chinese bumblebees are extremely rare [35]. To fill this knowledge gap, we provided the first comprehensive survey of the pathogens and parasites of bumblebees from different regions of China. We characterized the pathogen occurrences and communities using PCR detection. We then used a joint species distribution modeling framework [36] to disentangle the effects of host species and habitat/environmental covariates on pathogen co-occurrence patterns. Finally, we used the same approach to detect possible biotic community assembly processes from the pathogen–pathogen association matrices, while considering host variation, host species, and environmental variation.

We aim to answer the following questions: (1) Does the pathogen composition show a random pattern, or do some pathogens co-occur more or less often than what would be expected based on their frequencies? (2) What is the main factor affecting the pathogen community composition? (3) After accounting for these effects, is there evidence of pathogen co-occurrence patterns across all sampled bumblebee species that are indicative of competitive or facilitative pathogen interactions?

## 2. Materials and Methods

### 2.1. Field Sampling of Bombus Species

A total of 771 native bumblebees of 22 major species which are abundant in China were collected while foraging on flowers between July and October 2020 from 20 locations (Figure 1; more details of the sampling effects can be found in the Appendix A). It is important to note that we did not specifically characterize whether individual bumblebees were from the same or different colonies. Colony associations were not assessed as part of the field sampling protocol. This approach allowed us to investigate individual-level pathogen dynamics within the context of the broader bumblebee populations. All the bumblebee individuals were preserved in liquid nitrogen before being transferred to a −80 °C freezer in the lab and then confirmed the species by morphological identification using a published key [37]. All the specimens were deposited in the collection of the Institute of Apiculture Research, Chinese Academy of Agricultural Sciences, Beijing, China.

### 2.2. Pathogen Detection

We used PCR to detect the presence and prevalence of 13 known bumblebee pathogens in our Chinese bumblebee collection: acute bee paralysis virus, black queen cell virus, deformed wing virus, chronic bee paralysis virus, Israeli acute paralysis virus, Sacbrood virus, *Apicystis bombi*, *Crithidia bombi*, *Locustacarus buchner*, *Crithidia expoeki*, *Nosema bombi*, *Nosema apis*, and *Nosema ceranae*.

For each bumblebee, the abdomen was designated for total RNA/DNA extraction utilizing a specific RNA/DNA isolation kit (Huayueyang, Beijing, China) as per the instructions provided by the manufacturer. Once extracted, the RNA and DNA samples were immediately frozen and stored at −80 °C. Diagnostic PCRs were then executed for both the control group and the sampled bumblebees using specialized primers listed in the Appendix A. The extracted total RNA was converted into cDNA using the PrimeScript™ RT reagent Kit with gDNA Eraser (Takara, Shiga, Japan), following the guidelines supplied by the producer. PCR assays were conducted in 10 μL reaction volumes, which included 1 μL of DNA or cDNA, 5 μL of 2× Taq Plus PCR Master Mix (Tiangen, Beijing, China), 0.2 μL of each specific primer at a 10 μM concentration, and 3.6 μL of ddH2O. All PCR procedures included negative controls. The PCR cycles commenced with an initial 5 min phase at 95 °C, succeeded by 35 cycles at 95 °C for 30 s for denaturation, annealing at temperatures ranging from 46 to 60 °C for 30 s depending on the primer (refer to Appendix A), extension at 72 °C for 45 s, and a final elongation step lasting 10 min at 72 °C.

To validate the PCR results, we employed 2% agarose gel electrophoresis to visualize the produced PCR products and ensure they matched the expected fragment sizes. Positive PCR products were subsequently submitted for sequencing at the SinoGenoMax Company in Beijing, China. For each PCR product, sequencing was conducted in both forward and reverse directions. After obtaining the sequence pairs for each individual, we performed sequence trimming, alignment, and consensus sequence generation using BioEdit 7.0.0. To confirm the identity of the sequences, we conducted searches for each consensus sequence against the National Center for Biotechnology Information (NCBI) nucleotide database.

### 2.3. Hierarchical Modeling of Species Communities (HMSC) Framework

Although most research studies on bumblebee pathogens have focused on recording and documenting the presence and prevalence of pathogens [1], the use of joint species distribution models with community-scale species co-occurrence data and their responses to environmental variables can be a powerful tool to detect pathogen–pathogen associations and provide new insights into the determinants of the composition and assembly of pathogen communities [38,39]. Here, we applied a joint species distribution modeling framework, Hierarchical Modeling of Species Communities (HMSC) [36], which is a sophisticated multivariate Bayesian hierarchical generalized linear mixed model. The strength of the HMSC model is that it allows us to incorporate species occurrences, environmental covariates, species traits, and phylogenetic relationships from a multi-hierarchical study design with community assembly processes to examine the factors that govern the assembly of pathogen communities from the individual level to the population and community levels [40]. We used this approach to characterize the determinants of pathogen community structure as well as the pathogen–pathogen associations within Chinese bumblebees.

The response data matrix (Y matrix) comprises the presence/absence of 7 pathogens detected in 771 host individuals from 22 bumblebee species. We excluded Israeli acute paralysis virus and *Crithidia expoeki*, which were only detected once each in our study. We included the altitude of the sampling locality, annual mean temperature, and annual precipitation as fixed effects (X matrix). Additionally, we accounted for spatial variation at the community level by including a random effect with the GPS coordinates of the study sites (19 levels). We also considered the host characteristics by including the species of the host bumblebees, which added another community-level random effect with 22 levels. To investigate the pathogen–pathogen associations among bumblebees, we further introduced a community-level random effect for the individual hosts (771 levels). It is worth noticing that we only used broad-scale climate factors, such as annual temperature and precipitation, to infer the potential implication due to the lack of fine-scale climate data that align closely with bumblebee life cycles. Future research incorporating high-resolution weather data would undoubtedly yield insights that are currently obscured by the limitations of broader climatic measures.

We employed the HMSC (Hierarchical Modeling of Species Communities) model using the R-package Hmsc version 3.0 [40]. The model’s posterior distribution was estimated through four Markov Chain Monte Carlo (MCMC) chains, each running for 150,000 iterations, with the initial 50,000 iterations discarded as burn-in. To manage the data effectively, we thinned each chain by intervals of 100, resulting in 1000 representative posterior samples per chain and a cumulative total of 4000 samples. We assessed the model’s accuracy and convergence by analyzing species-specific Area Under the Curve (AUC) and Tjur’s Coefficient of Determination (R2) values. For model validation, we conducted a 5-fold cross-validation, dividing the complete set of 771 sampling communities into five subsets of equal size at random. A more comprehensive explanation of the HMSC model’s fitting process is available in the work by Tikhonov et al. (2020).

### 2.4. Determinants of Community Structure and Pathogen–Pathogen Associations

To evaluate the relative significance of each factor in determining the pathogen community structure in bumblebees, we conducted a variance partitioning analysis, separating the contribution of both the fixed and random effects integrated into the model. Subsequently, we compared the proportion of variance associated with each of these effects.

We characterized pathogen–pathogen associations by examining the species association matrix generated by HMSC. The values in this matrix ranged from −1 to 1, indicating the likelihood of two pathogens co-occurring (with a maximum of 1) or avoiding co-occurrence (reaching −1) when compared to random expectation [40]. The results could potentially indicate the ecological interactions among pathogens. For example, negative associations between pathogens suggest direct competition for resources, while positive associations could be evidence of facilitation of co-existence. We depicted the pathogen associations that had over 80% posterior support as either a negative or a positive interaction.

## 3. Results

### 3.1. Pathogens in Chinese Bumblebees

A total of 771 host individuals from 22 bumblebee species were sampled in 17 localities across China (Figure 1). Using real-time PCR with specific primers, we were able to detect and quantify the presence and prevalence of 13 pathogens in the 22 bumblebee species (Figure 1). Notably, we did not detect the chronic bee paralysis virus, *Nosema bombi*, *Nosema apis*, and *Nosema ceranae* in our study. Among the nine detected pathogens, the black queen cell virus, *Apicystis bombi*, *Crithidia bombi*, and *Locustacarus buchner* showed high presence and prevalence in almost all the sampled bumblebees (Figure 2). The remaining pathogens, including acute bee paralysis virus, Israeli acute paralysis virus, deformed wing virus, Sacbrood virus, and *Crithidia expoeki*, were only detected in a few species (Figure 2). Moreover, the infection rate of each pathogen varied across host species (Figure 2). For example, the highest infection rates of black queen cell virus, deformed wing virus, *Apicystis bombi*, *Crithidia bombi*, and *Locustacarus buchner* were found, respectively, in *B. braccatus* (71%, *n* = 28), *B. soroeensis* (80%, *n* = 5), *B. lantschouensis* (76%, *n* = 30), *B. longipes* (52%, *n* = 23), and *B. lantschouensis* (97%, *n* = 30) (Figure 2).

### 3.2. Pathogen Community Composition

The partitioning of variance among the explanatory variables, incorporating both fixed and random effects, revealed that host bumblebee species were the primary determinants of both the presence and the species composition of pathogens. Specifically, the host species accounted for over 40% of the total variance observed in the model (Figure 3). Climate conditions also affected bumblebee pathogen occurrence but to a lesser extent; annual mean temperature and annual precipitation together explained 8–17% of the variation (Figure 3).

### 3.3. Pathogen–Pathogen Associations

After accounting for factors such as host species, climate (temperature and precipitation), and geographical (locality and altitude) differences, the species association matrix reflects the potential associations between pathogens, with values ranging from −1 to 1. We found significant non-random positive associations among all four viruses, suggesting that they are more likely to co-exist in the same bumblebee host (Figure 4). Similarly, the three parasites, *C. bombi*, *A. bombi*, and *L. buchneri*, showed positive associations between each other, suggestive of potentially facilitatory associations (Figure 4). Moreover, significant non-random negative associations were found between viruses and parasites, indicating that viruses and parasites are less likely to co-exist in the same host, which may suggest a potential role for competition exclusion among them (Figure 4).

## 4. Discussion

Understanding the determinants of pathogen species composition and community assembly is a critical challenge for gaining insight into the dynamics of emerging infectious diseases, as well as their effects on biodiversity and ecosystem services [39,41,42,43,44]. In our study, we employed a joint species distribution modeling approach to analyze an extensive dataset of pathogen communities in Chinese bumblebees. Our findings underscore the significance of interspecific differences among bumblebee hosts in shaping the pathogen species composition and assembly. Furthermore, our results highlight the role of biotic interactions, such as competition, in determining pathogen community dynamics in Chinese bumblebees.

Overall, our result showed that most of the model-explained variation in pathogen occurrences was predominantly attributed to the host species (Figure 3), suggesting that the most important variable in predicting pathogen occurrences and community composition in bumblebees was host specificity. The importance of host species in bumblebee pathogen assembly could be related to the interspecific variation in host resistance, as both theoretical and experimental studies have shown that host resistance is a key determinant in the formation of pathogen communities [39,45,46]. Indeed, both pathogen occurrences and composition varied greatly among the 22 bumblebee species investigated in our study. In particular, *B. ignitus* and *B. braccatus* exhibited a higher infection prevalence and a greater diversity compared to other pathogen species (Figure 2). As the effects of abiotic variables, such as location and climate, have been controlled in our model, our findings suggest that the interspecific variation in pathogen resistance may play a significant role in the non-random distribution of pathogen occurrences in Chinese bumblebee species. In addition, the importance of the host species we observed in our study could also be related to the interspecific variation in host traits or host–pathogen co-evolution other than resistance [47,48]. For example, interspecific trait variation in bumblebees could indirectly affect pathogen occurrences through vector preference and the subsequent transmission mode [1]. Likewise, bumblebee species that are closely related share a common evolutionary history, and consequently, they also share common pathogens [48]. However, further studies investigating the details of these ecological interactions are needed to confirm which traits are involved. Moreover, by incorporating the effects of geography and climate in our model, we also found that abiotic variables have a potential role in affecting pathogen communities (Figure 3). Indeed, these variables often act as environmental filters, selecting particular pathogens from the regional pool based on their niche preferences, which, in turn, influences the composition of the local pathogen community within individual hosts [38,49,50]. Finally, it is worth noticing that cross-species infection dynamics may introduce a layer of complexity that can significantly impact bumblebee pathogen communities. For example, pathogens circulating in honeybee populations, where some of these pathogens are more frequent, can serve as potential sources of infection for bumblebees. These cross-species transmissions may, in some cases, exceed the influence of pathogen resistance mechanisms in bumblebees, thereby shaping the composition and prevalence of pathogens within bumblebee populations.

Apart from investigating the effect of host specificity, we examined the pathogen–pathogen associations among Chinese bumblebees by applying joint species distribution modeling and found evidence for both positive and negative pathogen–pathogen interactions. The associations between similar pathogens (viruses or non-viruses) tended to be positive, suggesting facilitatory interactions that could enhance their collective pathogenic impact. In contrast, the associations between different types of pathogens (viruses vs. non-viruses) were strongly negative, indicating competitive dynamics (Figure 4). Both positive and negative associations among parasites/pathogens have been reported in many studies, with the potential causes linked to host immune responses and competition [38,39,43,51,52,53]. For instance, the positive associations among similar pathogens in our study could be because the infection of one pathogen suppresses bumblebees’ immune response against similar pathogens, potentially facilitating their subsequent infections [38]. This could lead to a cascade effect where the presence of one pathogen paves the way for others in the same category, potentially overwhelming the host’s defenses. On the other hand, the strong within-host competition and priority effect among pathogens could potentially inhibit the infection of different types of pathogens, leading to the strong negative associations between virus and non-virus pathogens observed in our study [53]. Taken together, our results suggest that pathogen–pathogen associations also have an important role in driving pathogen co-occurrence patterns and community composition in bumblebees. It is worth noting that the pathogen–pathogen associations detected in our study should be considered hypotheses rather than facts. Future research linking these pathogen associations to variations in their traits, transmission modes, and evolutionary relationships will benefit our understanding of the community assembly of pathogens within bumblebees.

## 5. Conclusions

Bumblebees are among the most important pollinators on the planet, playing an essential role in both natural and agricultural ecosystems. They contribute significantly to maintaining biodiversity and crop production, supporting natural ecosystem functions, and the sustainability of human food resources. However, bumblebee populations have been recorded to decline worldwide due to global changes, such as habitat loss, climate change, pesticide use, and the spread of pathogens and diseases. Infectious diseases caused by multiple pathogen infections have been increasingly realized as a threat to the decline of bumblebees worldwide [1]. Understanding what regulates the assembly and composition of pathogen communities among bumblebee individuals and species can provide important implications for bumblebee conservation. By applying community ecology theories and modeling approach to comprehensive Chinese bumblebee pathogen community data, our study showed that interspecific variations in host species and biotic interactions among pathogens together determined the non-random distribution and structure of pathogen communities in Chinese bumblebees. Taken together, our research underscores the power of a comprehensive approach that combines field investigations of bumblebee pathogens with conceptual, statistical, and theoretical community ecology frameworks. This integrated approach has the potential to advance our understanding of the mechanisms and dynamics involved in multi-pathogen infectious diseases. Future research combining the above framework with large-scale manipulations of bumblebee–pathogen interactions will contribute to a deeper comprehension of the processes underlying disease emergence and control.

## Figures and Tables

**Figure 1 insects-14-00887-f001:**
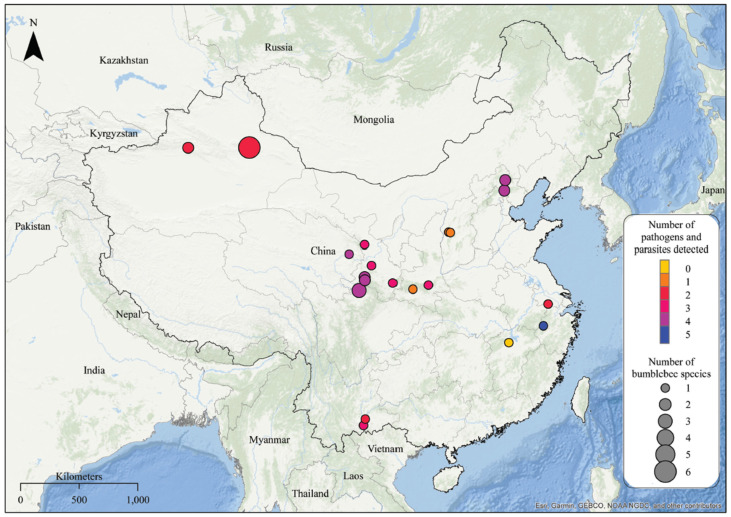
Bumblebee species sampling in mainland China for the detection of pathogens. The map displays the bumblebee sampling localities, with the size of the circle indicating the number of bumblebee species and the color of the circle representing the number of pathogens detected from that locality.

**Figure 2 insects-14-00887-f002:**
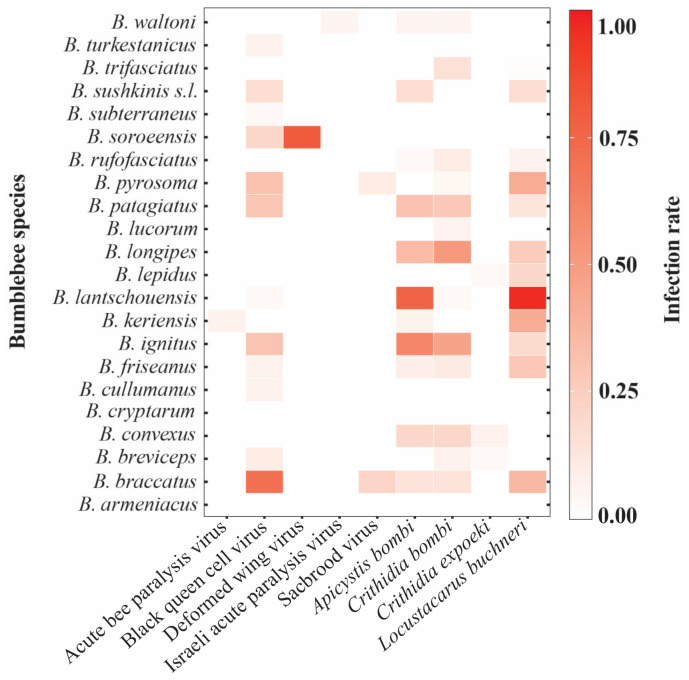
Pathogen occurrence and infection rate in the 22 bumblebee species sampled in China. The cells of the matrix are colored based on the infection rate of a given pathogen for a host species. Chronic bee paralysis virus, *N. bombi*, *N. apis*, and *N. ceranae* were not detected in our study.

**Figure 3 insects-14-00887-f003:**
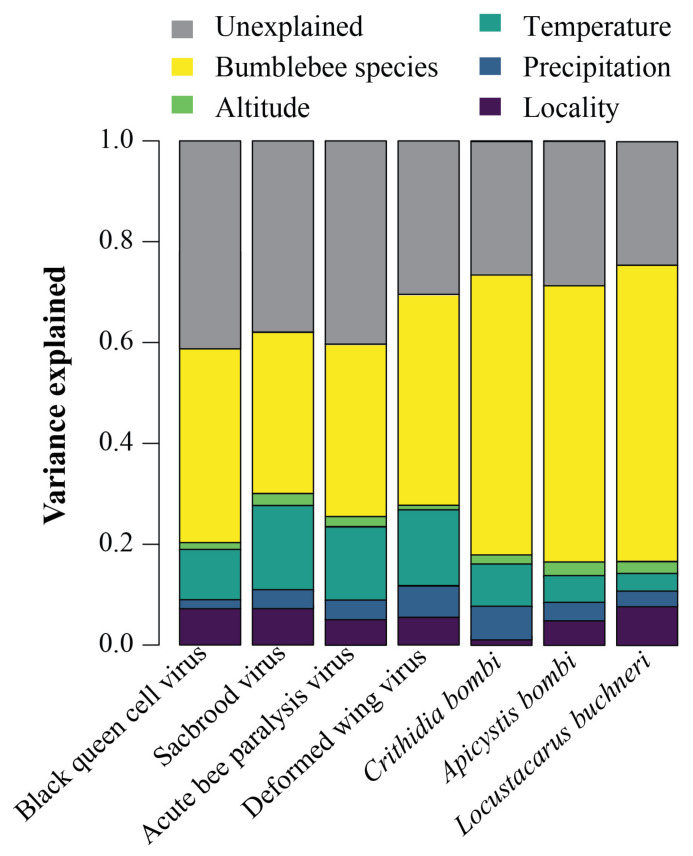
Joint species distribution modeling reveals the variance accounted for by each variable in relation to each pathogen (represented by columns). The species of the host bumblebees account for the largest proportion of the variance in the distribution of pathogen communities across individual bumblebees.

**Figure 4 insects-14-00887-f004:**
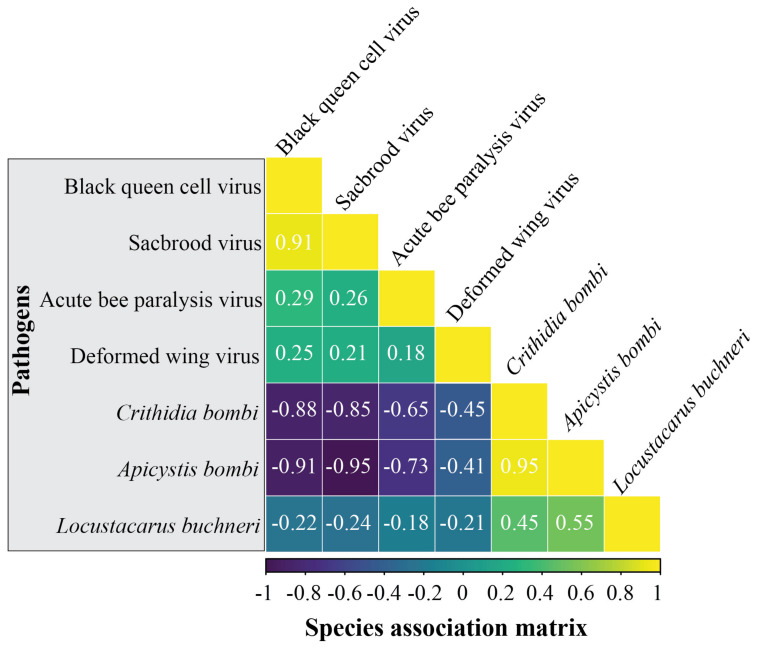
Pathogen−pathogen associations, supported by an 80% posterior probability according to Hierarchical Modeling of Species Communities (HMSC), are visually represented. Yellow (or blue) shading indicates pathogen pairs that are more inclined to co-occur (or less likely to do so). The numerical values signify the degree of species association.

## Data Availability

All data generated or analyzed during this study are included in this published article and its Appendix A.

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
