# Peer review of "Interspecific Host Variation and Biotic Interactions Drive Pathogen Community Assembly in Chinese Bumblebees"

_insects, 2023, doi:10.3390/insects14110887_

Round 1

Reviewer 1 Report

Comments and Suggestions for Authors

I would like some information on the number of samples of each species in each location. It is difficult to know the confidence in the results without this information. For example, perhaps you only had 2 samples of one species. There is no way of knowing.

If there was a way of showing on a map which species and pathogens were in which location I think that would help people to visualise your results in the context of location/environment/species/pathogen, even if it is just as supplementary material. Does the number of bumblebee species in a location correlate with the number of pathogens? While you have controlled for environmental variables, you cannot control for whether a species occurs in an area and whether species interact and spread pathogens to each other. For those species that had low pathogen prevalence was it low across all sites, or lower or higher when other species were present? I can see that in many of your locations you only samples 1-3 species but I can’t tell if these are questions you are able to ask as there is a lack of transparency in your results.

Line 24: you have not yet stated that there is a decline, only that bumblebees are important

Line 69: should this be ‘drift’

Line 186 and 187: remove ‘the’

Author Response

Reviewer 1’s comments

I would like some information on the number of samples of each species in each location. It is difficult to know the confidence in the results without this information. For example, perhaps you only had 2 samples of one species. There is no way of knowing.

Responses: Thank you very much for your comment. We have included our sampling data in the Supplementary Table S1.

If there was a way of showing on a map which species and pathogens were in which location I think that would help people to visualise your results in the context of location/environment/species/pathogen, even if it is just as supplementary material.

Responses: Thank you very much for this constructive suggestion to include a map illustrating the geographical distribution of the species and pathogens studied in our manuscript. We agree that including a geographical visualization would be an excellent addition to our manuscript. Unfortunately, it would be difficult to generate a clear map that including distribution of all bumblebee species and pathogens. To address this, we included our comprehensive sampling dataset in the Supplementary Table S1 which will be informative enough.

Does the number of bumblebee species in a location correlate with the number of pathogens? While you have controlled for environmental variables, you cannot control for whether a species occurs in an area and whether species interact and spread pathogens to each other. For those species that had low pathogen prevalence was it low across all sites, or lower or higher when other species were present? I can see that in many of your locations you only samples 1-3 species but I can’t tell if these are questions you are able to ask as there is a lack of transparency in your results.

Responses: We thank you for this comment. We understand your concern related to the issue of potential incomplete sampling. However, we do not think that would affect our results. In our HMSC modeling, we included host species as a random effect, which already accounted for any potential correlation between the number of bumblebee species and the number of pathogens.

Line 24: you have not yet stated that there is a decline, only that bumblebees are important

Responses: We have added one sentence as follows:

“However, recent reports of bumblebee decline have posed a significant threat to ecosystem stability.”

Line 69: should this be ‘drift’

Responses: changed

Line 186 and 187: remove ‘the’

Responses: changed

Reviewer 2 Report

Comments and Suggestions for Authors

It is an interesting paper dealing with the presence of various pathogens in Bombus species captured in China. It is well-written, and in my opinion, minor changes are needed for publication.

First of all, it is expected to use impersonal academic language, so please change the entire document according to this.

Regarding the results section and considering 22 bumblebee species were collected and studied and 13 pathogens investigated, I find it relevant to add a table showing the number of individuals of each species of bumblebee and the type of pathogens identified in each of them. I understand that part of this information is in Figure 2, but a table similar to what I suggest can facilitate understanding.

I understand that climate influence is important but maybe It should be used the weather parameters during the life cycle of the bumblebees more than the annual mean temperature or annual precipitation, which is a value so general for these types of studies. I do not if authors can use specific weather data (for example, daily means or monthly means) but in the cases that this is not possible, a discussion about the influence of weather parameters during the life cycle should be included.

Comment in lines 249-251, should be explained in more detail. Many factors that are not considered in the study such as cross-coinfections with other insects like, for example, Apis mellifera (in which some of these pathogens are very frequent) are not controlled during the study and can be a source of interaction between pathogens and bumblebees. Even more important than pathogen resistance.

In some cases, a large percentage of the variance of the data I unexplained (see figure 3). Please provide an explanation and discuss other factors that are not controlled and can influence the results.

I hope my comments help authors to improve their work.

Author Response

Reviewer 2’s comments

It is an interesting paper dealing with the presence of various pathogens in Bombus species captured in China. It is well-written, and in my opinion, minor changes are needed for publication.

Responses: Thank you very much.

First of all, it is expected to use impersonal academic language, so please change the entire document according to this.

Responses: changed

Regarding the results section and considering 22 bumblebee species were collected and studied and 13 pathogens investigated, I find it relevant to add a table showing the number of individuals of each species of bumblebee and the type of pathogens identified in each of them. I understand that part of this information is in Figure 2, but a table similar to what I suggest can facilitate understanding.

Responses: Thank you for your constructive comment. We understand the importance of clearly presenting the data, and we have included sampling data in the supplementary.

I understand that climate influence is important but maybe It should be used the weather parameters during the life cycle of the bumblebees more than the annual mean temperature or annual precipitation, which is a value so general for these types of studies. I do not know if authors can use specific weather data (for example, daily means or monthly means), but in the cases that this is not possible, a discussion about the influence of weather parameters during the life cycle should be included.

Responses: We appreciate this thoughtful comment. We agree that using specific weather parameters, particularly those aligned with the life cycle of bumblebees, can provide more nuanced insights into the relationship between climate and bumblebee pathogen dynamics. Unfortunately, the daily or monthly weather data for the relevant periods and locations of our study are not available after careful investigation.

While our current study utilized annual mean temperature and annual precipitation as broad-scale climate indicators, we acknowledge the potential value of incorporating more fine-scale weather data. We have added the following sentences in the method section:

“It is worth noticing that we only used broad-scale climate factors such as annual temperature and precipitation to infer the potential implication due to the lack of fine-scale climate data that align closely with bumblebee life cycles. Future research to incorporate high-resolution weather data, which would undoubtedly yield insights that are currently obscured by the limitations of broader climatic measures”.

Comment in lines 249-251, should be explained in more detail. Many factors that are not considered in the study such as cross-coinfections with other insects like, for example, Apis mellifera (in which some of these pathogens are very frequent) are not controlled during the study and can be a source of interaction between pathogens and bumblebees. Even more important than pathogen resistance.

Responses: Thank you for the insightful comments regarding the potential influence of cross-infections, especially from species such as Apis mellifera, on pathogen dynamics within bumblebee populations. We acknowledge that the interactions between different pollinator species and the transmission of pathogens amongst them represent a significant aspect of the ecological web within which bumblebees exist.

In our study, we focused on direct pathogen detection and quantification within bumblebee individuals and did not account for the indirect effects of pathogen spillover from other insects. We recognize that such cross-infections could indeed play a role in shaping pathogen communities within bumblebees and that the absence of this data is a limitation of our study. To address this, we have added the following sentences in the discussion:

“Finally, it is worth noticing that Cross-species infection dynamics may introduce a layer of complexity that can significantly impact bumblebee pathogen communities. For example, pathogens circulating in honeybee populations, where some of these pathogens are more frequent, can serve as potential sources of infection for bumblebees. These cross-species transmissions may, in some cases, exceed the influence of pathogen resistance mechanisms in bumblebees, thereby shaping the composition and prevalence of pathogens within bumblebee populations.”

In some cases, a large percentage of the variance of the data I unexplained (see figure 3). Please provide an explanation and discuss other factors that are not controlled and can influence the results.

Responses: Thank you for your comment. Like many other ecological studies, our models have a large portion of unexplained variance due to the complex and interconnected nature of ecological systems. The most likely reason in our study is the hidden variables. For example, there may be some unmeasured variables that can impact the occurrence and the composition of pathogens in bumblebees.

Reviewer 3 Report

Comments and Suggestions for Authors

Bumblebees are important pollinators for both wild plants and agricultural crops. However, the population of bumblebees have been experiencing a remarkable decline worldwide, which is driven by multiple factors including infectious disease caused by pathogens and parasites. Therefore, it is critical to understand how the pathogen communities are assembled. This study has focused on this question and disentangled host interspecific variations and biotic interactions likely determining the occurrence of pathogens in Chinese bumblebees. The experiments were carefully performed and the data were properly analyzed. However, the writing of the manuscript needs to be improved and a few minor concerns listed below have to be solved:

L43. Should be “genus: bombus” to make a clearer.

L70. Better use “among pathogens” instead of “in pathogens”

L72-78. Host resistance involves the interaction between host and pathogens. So what is the difference between host resistance and biotic interactions?

L89-96. Use past tense to describe what you did.

L104. Did you characterize if individual bumblebees were from the same colony or not? Bees from the same colony may share pathogen species by social interaction, which would affect the investigation of presence and prevalence of pathogens. Some details on the field sampling are needed.

L107. Should be “transferred”.

L138-143. Move this long sentence to Introduction.

L143. Use past tense for “apply”.

L147. Use past tense as well.

L150. Use past tense.

L153. Use past tense.

L182-183. I see the majority of bumblebees came from North China. Why not include more samples from South China?

L186. Should be 9 pathogens.

L229. Blue or red? Double-check it.

L287-288. This sentence is logically confusing since we cannot say something is important in the decline of bees. A revision is needed.

Comments on the Quality of English Language

The quality of English is generally high, however, there are still some issues needed to be solved.

Author Response

Reviewer 3’s comments

Bumblebees are important pollinators for both wild plants and agricultural crops. However, the population of bumblebees have been experiencing a remarkable decline worldwide, which is driven by multiple factors including infectious disease caused by pathogens and parasites. Therefore, it is critical to understand how the pathogen communities are assembled. This study has focused on this question and disentangled host interspecific variations and biotic interactions likely determining the occurrence of pathogens in Chinese bumblebees. The experiments were carefully performed and the data were properly analyzed. However, the writing of the manuscript needs to be improved and a few minor concerns listed below have to be solved:

Responses: Thank you for your efforts to improve our MS.

L43. Should be “genus: bombus” to make a clearer.

Responses: changed

L70. Better use “among pathogens” instead of “in pathogens”

Responses: changed

L72-78. Host resistance involves the interaction between host and pathogens. So what is the difference between host resistance and biotic interactions?

Responses: The biotic interactions here refer to the interactions between different bumblebee species. We have modified the original sentence as follows:

“Another deterministic hypothesis suggests that the interaction dynamics between bumblebee hosts play a significant role in shaping pathogen communities.”

L89-96. Use past tense to describe what you did.

Responses: changed

L104. Did you characterize if individual bumblebees were from the same colony or not? Bees from the same colony may share pathogen species by social interaction, which would affect the investigation of presence and prevalence of pathogens. Some details on the field sampling are needed.

Responses: We appreciate the reviewer's question regarding the characterization of individual bumblebees' colony associations in our study. In our study, we did not specifically characterize whether individual bumblebees were from the same colony or different colonies. Our sampling approach focused on the collection of bumblebees from various locations. While we acknowledge that colony-level interactions can influence pathogen transmission dynamics, it would be very difficult to characterize each bumblebee individual’s colony association.

We have added a clarification in the methodology section to explicitly state that colony associations were not assessed during our field sampling.

“It is important to note that we did not specifically characterize whether individual bumblebees were from the same or different colonies. Colony associations were not assessed as part of the field sampling protocol. This approach allowed us to investigate the individual-level pathogen dynamics within the context of the broader bumblebee populations.”

L107. Should be “transferred”.

Responses: changed

L138-143. Move this long sentence to Introduction.

Responses: We appreciate your suggestion. However, we think it is important to leave that sentence in the current section so that we can emphasize the importance of using joint species distribution models to detect pathogen-pathogen associations and provide new insights into the determinants of composition and assembly of pathogen communities.

L143. Use past tense for “apply”.

Responses: changed

L147. Use past tense as well.

Responses: changed

L150. Use past tense.

Responses: changed

L153. Use past tense.

Responses: changed

L182-183. I see the majority of bumblebees came from North China. Why not include more samples from South China?

Responses: Thank you for your comment regarding our bumblebee sampling. We focused on North China because Chinese bumble species are mostly distributed in North China. While we noticed the value of adding more examples from South China would be beneficial, we were unable to conduct such sampling due to limited resources and time.

L186. Should be 9 pathogens.

Responses: changed

L229. Blue or red? Double-check it.

Responses: changed

L287-288. This sentence is logically confusing since we cannot say something is important in the decline of bees. A revision is needed.

Responses: We have modified the sentence as follows:

“Infectious diseases caused by multiple pathogen infections have become a threat to the decline of bumblebees worldwide.”

Round 2

Reviewer 1 Report

Comments and Suggestions for Authors

Some summary tables would have been good in the supplementary material but can be accepted without